# *TP53* Mutation Analysis in Gastric Cancer and Clinical Outcomes of Patients with Metastatic Disease Treated with Ramucirumab/Paclitaxel or Standard Chemotherapy

**DOI:** 10.3390/cancers12082049

**Published:** 2020-07-24

**Authors:** Francesco Graziano, Nicholas W. Fischer, Irene Bagaloni, Maria Di Bartolomeo, Sara Lonardi, Bruno Vincenzi, Giuseppe Perrone, Lorenzo Fornaro, Elena Ongaro, Giuseppe Aprile, Renato Bisonni, Michele Prisciandaro, David Malkin, Jean Gariépy, Matteo Fassan, Fotios Loupakis, Donatella Sarti, Michela Del Prete, Vincenzo Catalano, Paolo Alessandroni, Mauro Magnani, Annamaria Ruzzo

**Affiliations:** 1Medical Oncology Unit, Azienda, Ospedali Riuniti Marche Nord, 61121 Pesaro, Italy; d.sarti@fastwebnet.it (D.S.); catalano_v@yahoo.it (V.C.); paolo.alessandroni@ospedalimarchenord.it (P.A.); 2Genetics & Genome Biology Program, The Hospital for Sick Children, Toronto, ON M5G 0A4, Canada; nick.fischer@mail.utoronto.ca (N.W.F.); david.malkin@sickkids.ca (D.M.); 3Department of Biomolecular Sciences, Università degli Studi di Urbino, 61032 Fano, Italy; irene.bagaloni@uniurb.it (I.B.); mauro.magnani@uniurb.it (M.M.); 4Department of Medical Oncology, Istituto Nazionale dei Tumori di Milano, 20133 Milan, Italy; Maria.DiBartolomeo@istitutotumori.mi.it (M.D.B.); michele.prisciandaro@istitutotumori.mi.it (M.P.); 5Veneto Institute of Oncology IOV–IRCCS, 35128 Padova, Italy; sara.lonardi@iov.veneto.it (S.L.); matteo.fassan@unipd.it (M.F.); fotios.loupakis@iov.veneto.it (F.L.); 6Department of Oncology, Campus Bio-Medico University, 00128 Rome, Italy; B.Vincenzi@unicampus.it (B.V.); g.perrone@unicampus.it (G.P.); 7Unit of Medical Oncology 2, Azienda Ospedaliero-Universitaria Pisana, 56126 Pisa, Italy; lorenzo.fornaro@gmail.com; 8Department of Oncology, University and General Hospital, 33100 Udine, Italy; eleongaro@gmail.com; 9Unit of Medical Oncology and Cancer Prevention, Department of Medical Oncology, Centro di Riferimento Oncologico di Aviano (CRO) IRCCS, 33081 Aviano, Italy; 10Department of Oncology, San Bortolo General Hospital, 36100 Vicenza, Italy; giuseppe.aprile@aulss8.veneto.it; 11Medical Oncology Unit, Hospital of Fermo, 63900 Fermo, Italy; renato.bisonni@sanita.marche.it (R.B.); micheladelprete@gmail.com (M.D.P.); 12Division of Hematology-Oncology, Department of Pediatrics, The Hospital for Sick Children, Toronto, ON M5G 1X8, Canada; 13Department of Medical Biophysics, University of Toronto, Toronto, ON M5G 1L7, Canada; jean.gariepy@utoronto.ca; 14Physical Sciences, Sunnybrook Research Institute, Toronto, ON M4N 3M5, Canada

**Keywords:** gastric cancer, *TP53*, Ramucirumab, Paclitaxel, angiogenesis

## Abstract

Loss of p53 promotes vascular endothelial growth factor (VEGF)-A up-regulation and the angiogenic potential of cancer cells. We investigated *TP53* somatic mutations in 110 primary gastric adenocarcinomas of two retrospective metastatic series including 48 patients treated with second-line Ramucirumab/Paclitaxel and 62 patients who received first-line chemotherapy with Cisplatin or Oxaliplatin plus 5-Fluorouracil. Missense mutations were classified by tumor protein p53 (*TP53*) mutant-specific residual transcriptional activity scores (*TP53*_RTAS_) and used to stratify patients into two groups: transcriptionally *TP53*_Active_ and *TP53*_Inactive_. The primary endpoint was overall survival (OS). An additional analysis was addressed to measure VEGF/VEGF receptor 2 (VEGFR2) expression levels in relation to the *TP53*_RTAS_. In the Ramucirumab/Paclitaxel group, 29/48 (60.4%) patients had *TP53* mutations. Ten patients with *TP53*_Inactive_ mutations showed better OS than carriers of other *TP53* mutations. This effect was retained in the multivariate model analysis (Hazard Ratio = 0.29, 95% confidence interval = 0.17–0.85, *p* = 0.02). In the chemotherapy group, 41/62 (66%) patients had *TP53* mutations, and the 11 carriers of *TP53*_Inactive_ mutations showed the worst OS (Hazard Ratio = 2.64, 95% confidence interval = 1.17–5.95, *p* = 0.02). VEGF-A mRNA expression levels were significantly increased in *TP53*_Inactive_ cases. Further studies are warranted to explore the effect of *TP53*_Inactive_ mutations in different anti-cancer regimens. This information would lead to new tailored therapy strategies for this lethal disease.

## 1. Introduction

Tumor protein p53 (*TP53*) is a multifunctional tumor suppressor gene that is intimately involved in the control of target genes that regulate “healthy” biological processes, including cell-cycle arrest, apoptosis, senescence, energy metabolism, and antioxidant defense to prevent tumorigenesis [1]. In recent years, several experimental and clinical studies have also indicated a role for *TP53* in the control of tumor angiogenesis [2]. This effect seems to be linked to cross-talk mechanisms between *TP53*, vascular endothelial growth factor (VEGF), and VEGF receptors.

A highly conserved and functional p53-binding site has been identified within the *VEGF* promoter and the p53 tumor suppressor downregulates VEGF expression [3]. Loss of *TP53* in tumor cells enhances HIF-1alpha levels and augments HIF-1-dependent transcriptional activation of the *VEGF* gene in response to hypoxia [4]. *TP53*-deficient cancer cells were found to produce reactive oxygen species, which activated fibroblasts to mediate angiogenesis by VEGF both in-vivo and in-vitro [5]. The transcription factor E2F1 showed regulation of angiogenic activity via p53-dependent transcriptional control of VEGF expression [6]. In experimental models, mutant *TP53* can up-regulate the transcription of VEGF receptor 2 (VEGFR2) by promoter remodeling [7]. These molecular mechanisms may explain analyses of human cancer tissues that have reported significant increases in VEGF expression levels in the presence of *TP53* mutations [8,9,10]. Interestingly, in a large pan-cancer study [9], the association between VEGF up-regulation and *TP53* mutants remained independent of *HIF-1* and *MDM2* overexpression. This translational background explains recent clinical findings in advanced cancer patients who had improved responses and survival outcomes after VEGF/VEGF receptor (VEGFR) inhibitor therapy mostly in tumors harboring a *TP53* mutation [11,12,13,14,15].

The concept that *TP53* alterations may represent a favorable biomarker for treating patients with anti-angiogenesis agents contrasts with previous findings from standard chemotherapy studies, where *TP53* dysregulation was generally associated with poor clinical outcomes [16]. However, this is not surprising considering the multiple and widespread roles of *TP53* and the prevalence of p53-associated mechanisms of chemoresistance [16].

Despite decades of research, the analysis of the *TP53* status for predictive purposes in cancer therapy has not been implemented in routine clinical practice yet. Major limitations concern the lack of standardized methods for defining the *TP53* status in tumor samples. Mutational analysis is more reliable than immunohistochemistry in solid tumors, but somatic *TP53* mutations cannot be considered a homogeneous group inducing an on/off effect [1]. The majority of *TP53* mutations occurring in human solid neoplasms are missense mutations with a large gradient of functional consequences [1]. Missense *TP53* mutations can be classified for clinical purposes by considering the residual transcriptional activity score (*TP53_RTAS_*) [17], derived from the results of a site-directed mutagenesis technique and yeast-based functional assay [18].

Gastric cancer ranks among the most frequently *TP53*-mutated solid tumors [19], and in recent years, the anti-VEGFR2 inhibitor Ramucirumab coupled with Paclitaxel has become standard second-line systemic therapy in this lethal disease [20]. Unfortunately, the magnitude and the duration of the survival gain in Ramucirumab/Paclitaxel treated patients are limited and the discovery of predictive markers would improve the selection of patients and allow the adoption of novel combination therapies [21].

This background prompted us to plan a translational study in patients with metastatic gastric cancer treated with Ramucirumab/Paclitaxel including the analysis of *TP53* mutations and *TP53_RTAS_* in their tumor samples. The association between the mutant *TP53* functional status and survival outcome was assessed and overall patient survival was the primary endpoint of the study. To better characterize the predictive impact of *TP53* mutations, an additional retrospective cohort of patients treated with standard chemotherapy for advanced disease was included in the study.

## 2. Results

The overall study population consisted of 110 gastric cancer patients whose primary tumors were analyzed for *TP53* mutations. The study group included 48 cases who underwent second-line Ramucirumab/Paclitaxel. In the control group, 62 patients were treated with standard first-line chemotherapy with a 5-Fluorouracil and a platinum compound (Cisplatin or Oxaliplatin).

### 2.1. TP53 Analysis in Primary Gastric Tumors

As shown in Table 1, 61 *TP53* mutations were detected in total, including 47 missense mutations (77%), 7 nonsense mutations (11.4%), 4 frameshift mutations (6.6%), 2 splice site mutations (3.3%), and 1 in-frame deletion (1.7%). Some “hot-spot” missense mutations occurred in more than one patient: p.R282W and p.G244D in two cases, p.R283H in three cases, p.R273C in five cases. Four patients showed a combination of two or more *TP53* mutations in their tumor samples. Overall, 70 out of 110 patients showed tumor samples positive for *TP53* mutations (63.6%). The distribution of *TP53* mutations (any type) according to clinical and pathological characteristics of patients and tumors is shown in Table 2. No significant association was found except for a prevalence of *TP53* mutations in intestinal-type gastric cancer according to Lauren’s classification (Table 2).

### 2.2. Classification of TP53 Mutations and Study Groups

Results of the residual transcriptional activity score (RTAS) analysis for missense mutations (*TP53_RTAS_*) are listed in Table 1. *TP53_Inactive_* missense mutations were found in 10 patients in the Ramucirumab/Paclitaxel group and 11 patients in the chemotherapy control group. The remaining 49 *TP53* mutation-positive patients were classified as carriers of a *TP53_Active_* missense mutation and carriers of non-missense mutations (nonsense, frameshift, splice-site, and in-frame deletions). *TP53_Active_* missense mutation carriers were in 13 cases in the Ramucirumab/Paclitaxel group and 25 cases in the chemotherapy control group. Non-missense mutations carriers totaled 5 in the Ramucirumab/Paclitaxel group and 6 in the chemotherapy control group.

### 2.3. Ramucirumab/Paclitaxel Second-Line Therapy and TP53 Analysis

In the 48 patients of the study group, the results of the second-line therapy showed a 20.8% overall response rate (10 patients with a partial response) and a median overall survival (OS) time of 8.4 months (5–8.8 months 95% CI). No significant association was detected between *TP53* mutations and tumor response. Partial responses occurred in three patients with *TP53_Inactive_* missense mutations, in two patients with *TP53* non-missense mutations, and in five patients with wild-type *TP53_RTAS_* status.

Median OS times were: 9.5 months (9.0–10.7 months 95% CI) in carriers of *TP53_Inactive_* missense mutations; 8.6 months (5.9–9.9 months 95% CI) in carriers of other *TP53* mutations; 6.0 months (3.2–8.5 months 95% CI) in carriers of *TP53_Active_* missense mutations; 4.5 months (4.1–8.2 months 95% CI) in patients without *TP53* mutations. A significant difference was observed between the survival curves of the four groups using the log–rank test (Figure 1). The analysis of hazard ratios with 95% CIs indicates the survival advantage of carriers of *TP53_Inactive_* missense mutations over other groups except for carriers of other *TP53* mutations (Figure 1). The favorable effect of the *TP53_Inactive_* mutational status was retained in the multivariate model (Figure 2).

### 2.4. Standard First-Line Chemotherapy and TP53 Analysis

In the 62 patients of the control group, the results of the first-line chemotherapy showed a 51.6% overall response rate (28 partial responses and 4 complete responses). The median OS time was 9 months (95% Cls = 8–10.2 months). No significant association was detected between *TP53* mutations and tumor response. Partial responses occurred in 5 patients (45%) with *TP53_Inactive_* missense mutations, in 13 patients (52%) with *TP53_Active_* missense mutations, in 2 patients (40%) with *TP53* non-missense mutations, and in 8 patients (38%) without *TP53* mutations. Complete responses were observed in one patient in each of the four groups. Median OS times were: 8 months (4.3–9.0 months 95% CI) in carriers of *TP53_Inactive_* missense mutations; 8 months (8.4–14.7 months 95% CI) in carriers of other *TP53* mutations; 8.5 months (5.7–10 months 95% CI) in carriers of *TP53_Active_* missense mutations; 10.6 months (8.4–14.7 months 95% CI) in patients without *TP53* mutations. A comparison of the survival curves using the log–rank test showed significant differences between the four groups (Figure 3).

The analysis of hazard ratios with 95% CIs reveals a detrimental effect of the *TP53_Inactive_* missense mutations status in comparison to patients without *TP53* mutations (Figure 3). The adverse effect of the *TP53_Inactive_* mutational status was retained in the multivariate model (Figure 2).

### 2.5. VEGF/VEGFR Analysis and TP53 Mutational Status in Gastric Cancer Tissues

Since Ramucirumab is a VEGFR2 antagonist that blocks the binding of VEGF-A, VEGF-C, and VEGF-D, we analyzed the mRNA expression and copy number alterations of these genes in gastric adenocarcinomas. *VEGF-A* gene gain was significantly more frequent in tumors with *TP53_Inactive_* mutations (58.1%) as compared to tumors with *TP53_Active_* mutations (35.7%) or wild-type p53 (13.4%) (*p* = 0.019 and *p* < 0.0001, respectively). Importantly, *VEGF-A* mRNA expression was correspondingly higher in the *TP53_Inactive_* group as compared to tumors with *TP53_Active_* or wild-type p53 (*p* = 0.047 and *p* = 0.0039, respectively). While no differences in the gene loss of *VEGF-A* were observed between these groups, the deletion of *VEGF-C* and *VEGFR2* occurred less often in the wild-type p53 group as compared to the *TP53* mutation subgroups (*p* < 0.01 and *p* < 0.0001, respectively), although this did not translate to significant differences in mRNA expression levels. Loss of *VEGF-D* occurred most frequently in the *TP53_Inactive_* group (32.6%), with significantly fewer deletion events in wild-type p53 tumors (*p* < 0.0001), however, there were no differences in mRNA expression levels (Figure 4). Together, these findings support a mechanism exclusive to tumors with transcriptionally inactive p53 mutants, indicating a reliance on increased VEGF-A production to drive tumorigenesis.

## 3. Discussion

The results of this study support the hypothesis that *TP53* may be a valuable biomarker that can identify metastatic gastric cancer patients with the greatest benefit from an anti-angiogenic, anti-VEGFR2 systemic therapy. Importantly, the positive therapeutic effect, being associated with a specific group of transcriptionally inactive *TP53* missense mutations (*TP53_RTAS_* < 1%) would simplify the development of a genetic test for further investigations, and hopefully, for routine clinical practice. This finding contributes to a mounting body of evidence linking *TP53* mutational status to anti-angiogenic treatment clinical outcomes in patients with advanced cancers [11,12,13,14,15].

So far, the loss of function of the *TP53* tumor suppressor gene has been considered an unfavorable prognostic feature in patients with solid tumors [22]. Uncontrolled cell-cycle regulation, senescence, metabolism, and apoptosis in *TP53* “null” neoplasms may explain this association [22]. However, the clinical impact of *TP53* dysregulation may vary in patients undergoing anti-cancer systemic therapies, which could depend on differences in the mechanisms of action of anti-cancer agents [16]. Pre-clinical and translational studies have found links between *TP53* loss of function and resistance to DNA damaging agents like platinum compounds and anthracyclines [16]. Conversely, tumors with loss of normal *TP53* function may be even more sensitive to anti-cancer agents like Paclitaxel that stabilizes tubulin polymerization resulting in the arrest of mitosis and the induction of *TP53*-independent apoptosis [23,24]. It has been also demonstrated that Paclitaxel, especially in fractionated regimens, exploits anti-angiogenic mechanisms of action [25] Together, these chemotherapy-related aspects, in addition to pre-clinical and clinical studies linking *TP53* mutations to the VEGF pathway [5,6,7,8] and anti-VEGF/VEGFR systemic therapies [9,10,11,12,13,14,15], contribute to explaining the favorable results of the Ramucirumab/Paclitaxel combination in metastatic gastric cancers harboring *TP53* mutations.

In the present study, we performed a combined analysis of *TP53_RTAS_* missense mutations and VEGF-A and VEGFR2 expression levels in gastric adenocarcinoma tumor tissue samples. The results indicate a significant VEGF-A up-regulation in tumor samples with *TP53_Inactive_* and unmodified VEGFR2 expression. These results parallel findings in previous analyses [3,4,5,6,7]. In a large pan-cancer cohort of 7525 samples, Li AM et al. [9] demonstrated up-regulated VEGF-A transcript levels in tumors with *TP53* mutations, particularly in adenocarcinomas, regardless of their organ of origin, while VEGFR2 expression levels were not significantly modified by *TP53* mutational status or reduced in squamous carcinomas. Since VEGF-A is considered the most potent angiogenic ligand and it exhibits the highest binding affinity for VEGFR2 [26], it is plausible that VEGF-A up-regulation is a major mechanism underlying the positive clinical impact of *TP53* mutants on anti-VEGF/VEGFR2 therapies.

Intriguingly, additional mechanisms may also explain the positive clinical interaction between chemotherapy, anti-angiogenics, and *TP53* status. In a translational analysis from a randomized trial in endometrial cancer, a remarkable survival benefit was found in the bevacizumab/chemotherapy arm in the presence of *TP53* mutations causing loss of function or “null” phenotype [14]. Results from cell models suggested a mechanism of synthetic lethality derived from the effects of agents like bevacizumab to abrogate cell cycle checkpoints in the absence of p53 by blocking signaling downstream of tyrosine kinases [14]. This causes the premature entry of cancer cells into vulnerable phases of the cell cycle where chemotherapy agents are most effective.

The majority of somatic *TP53* mutations detected in human cancers are missense mutations [1,2]. These mutations, which arise from a point mutation in a single nucleotide, can result in amino acid changes that can lead to highly variable degrees of functional consequences. For example, an amino acid may be replaced by another amino acid with very similar chemical properties, resulting in a protein that still functions normally. In contrast, some amino acid changes may cause greater dysfunction or non-functional protein products. To overcome difficulties in the interpretation of *TP53* mutational analysis, we adopted a functional classification of *TP53* missense mutations based on a transcriptional activity score as the result of a site-directed mutagenesis technique and yeast-based functional assay [17,18]. Tumors harboring *TP53_Inactive_* missense mutations showed the longest survival time and the greatest benefit from the anti-VEGFR2 Ramucirumab/Paclitaxel systemic therapy. The analysis of survival curves suggests that tumors with *TP53* non-missense mutations may also obtain some survival benefit from Ramucirumab/Paclitaxel (non-significant 52% risk reduction in the comparison with the wild-type group). Notably, *TP53* non-missense mutants producing the loss of the protein product do not display specific properties of some missense mutations (i.e., hotspot mutants) with augmented oncogenic potential. This effect may be caused by their capacity to impair the wild-type allele (dominant-negative effects) and/or by specific gain-of-function effects [25,26]. Many missense *TP53* mutants are expressed as stable proteins that exert dominant-negative effects by interfering with the remaining wild-type p53 protein copies through the formation of hetero-tetramers. A “prion-like” effect of some p53 mutants has also been shown to inactive wild-type p53 in vitro by forcing the wild-type protein to adopt a denatured, mutant-like conformation [2,27]. As a result of the gain-of-function effect, some *TP53* missense mutants were found to promote tumor angiogenic pathways, whereas *TP53* deletion or truncating events did not [28,29].

Our analysis of *TP53* mutations in relation to the clinical characteristics and pathological features of gastric adenocarcinomas in the present study population is supported by pivotal studies on the molecular classification of gastric cancer [30]. *TP53* mutations characterize the most common of four molecular subtypes of gastric adenocarcinomas, defined by chromosomal instability. This genomically unstable subtype is associated with an intestinal histotype according to Lauren’s classification, and a homogenous distribution along the different gastric sites.

## 4. Materials and Methods

The study group consisted of metastatic gastric cancer patients who received second-line systemic therapy with Ramucirumab 8 mg/kg (given on day 1 and 15) and Paclitaxel 80 mg/m^2^ (given on day 1, 8, and 15), both administered intravenously every 28 days. The cohort study was retrospective and performed among participating Institutions in the RAMoss analysis [31], which retrospectively evaluated the safety and efficacy of Ramucirumab among Italian patients failing first-line treatment for advanced gastric cancer.

The control group comprised metastatic gastric cancer patients who underwent Cisplatin or Oxaliplatin plus 5-Fluorouracil systemic chemotherapy. This retrospective cohort was implemented from consecutive cases included in a large three-Institution database [32]. In both cohorts, the study inclusion required the availability of primary tumor tissue samples. The study was performed in accordance with the reporting recommendations for tumor marker prognostic studies (REMARK) guidelines [33]. All patient information and pathology materials were collected under a protocol approved by the Regional Ethical Committee (the protocol number is 2016-0374MN).

### 4.1. Samples and Nucleic Acids Extraction

A sample of 4–6 10-μm sections from formalin-fixed, paraffin-embedded specimens were obtained from patient tumors and matched normal tissues. Before cutting sections for total nucleic acid isolation, an additional slide was prepared for hematoxylin-eosin staining and the pathologists identified representative areas with an almost complete representation of tumor infiltration. Tissues were micro-dissected and placed in a 1.5 mL reaction tube containing 1 mL xylene to remove paraffin. DNA was extracted using the RecoverAll^TM^ Multi-Sample RNA/DNA Isolation Workflow (Invitrogen^TM^ by Thermo Fisher, Foster City, CA, USA) according to the manufacturer’s instructions. DNA concentration and purity were measured using the NanoDrop 1000 spectrophotometer (Nanodrop Technologies, Rockland, DE, USA).

### 4.2. Amplicons Library Preparation and Next-Generation Sequencing (NGS) for TP53 Analysis

A custom panel (IAD_119861) including the *TP53* gene coding and UTR regions was designed using the Ion AmpliSeq™ Designer software (Thermo Fisher, Foster City, CA, USA). The panel was made up of 35 amplicons and ensured 82% of coverage for DNA from formalin-fixed paraffin-embedded (FFPE) tissues. Library preparation was performed using the Ion AmpliSeq Library Kit Plus according to the manufacturer’s instructions. Libraries were generated using 40 ng of DNA from tumor FFPE sections and indexed using the Ion Xpress Barcode Adapter Kit. Library purification was carried out using the AMPure^TM^ XP Reagent (Beckman Coulter, Brea, CA, USA) on the DynaMag^TM^-2 Magnet. Qubit^TM^ 4 Fluorometer (Invitrogen^TM^, by Thermo Fisher, Foster City, CA, USA) was used to quantify amplicons libraries. After dilution of all samples at 100 pM, libraries were pooled for emulsion PCR on the Ion OneTouch™ 2 instrument, using the Ion S5™ Template OT2 kit, according to the manufacturer’s instructions. The Ion Sphere™ Particles were enriched using the Ion OneTouch™ Enrichment System and the template was sequenced on the Ion Torrent S5 platform using the Ion 540^TM^ Chip (cat.no.A27766) following the manufacturer’s instruction. All of these instruments and reagents were supplied by Thermo Fisher (Foster City, CA, USA). Read alignment was performed using hg19 (GRCh37) as the reference genome. Variant call files (VCF) were generated by the Variant Caller v.5 plugin preinstalled in the Torrent Suite and analyzed with the Ion Reporter™ software (Thermo Fisher, Foster City, CA, USA). BAM files were also manually checked on IGV (Integrative Genomics Viewer) [34].

### 4.3. Classification of TP53 Mutations

Each *TP53* missense mutation was assigned a residual transcriptional activity score (*TP53_RTAS_*) according to the results of a site-directed mutagenesis technique and yeast-based functional assay [17,18]. The *TP53_RTAS_* represents the median transcriptional activity value measured across eight different p53-responsive elements. Based on these functional scores, *TP53* missense mutations were then divided into two distinct groups: *TP53_RTAS_* ≥ 1% and *TP53_RTAS_* < 1%. This categorization denotes a clear distinction between a transcriptionally inactive group (*TP53_Inactive_ =TP53_RTAS_* < 1%) versus a transcriptionally active group (*TP53_Active_ =TP53_RTAS_* ≥ 1%). Carriers of non-missense mutations including nonsense and frameshift mutations were merged into a third mutational group.

### 4.4. VEGF and VEGFR Analyses

A gastric adenocarcinoma dataset was collected from the TCGA Pan-Cancer Atlas (https://www.cancer.gov.tcga) for the analysis of mRNA expression, copy number alterations, and mutational data of genes of interest. Tumors with *TP53* gene sequencing were selected and those with more than one *TP53* alteration were excluded. Individual tumors were then assigned a *TP53* mutation-specific RTAS, sub-grouped based on the RTAS, and analyzed for the gene expression and copy gain or loss of *VEGF-A*, *VEGF-C*, *VEGF-D*, and *VEGFR2*.

### 4.5. Statistical Analysis

The primary endpoint of the study was the overall survival (OS) analysis in carriers of *TP53_Inactive_* mutations in the Ramucirumab/Paclitaxel study group. With 40 events and a 20% prevalence of the *TP53_Inactive_* mutational status, the scenario for sample size estimation would allow detection of a 66% reduced risk of death with a power of 80% and a two-sided type I error of 5%. In the Ramucirumab/Paclitaxel group, OS was calculated from the date of the first cycle of the second-line therapy to the date of death or last follow-up. In the chemotherapy control group, OS was calculated from the date of the first cycle of the first-line therapy to the date of death or last follow-up. The Kaplan–Meier method was used to estimate survival curves and the log–rank test was used to compare survival times between groups. A multivariate Cox proportional hazards model was adopted for adjusting according to clinical and pathological features. Patients achieving complete response or partial response and patients with stable disease or disease progression were evaluated according to the RECIST criteria and the overall response rate included patients with a complete response and partial response. Contingency tables were analyzed by the Chi-square test. All reported *p*-values were two-sided, and confidence intervals (CIs) were at the 95% level. A *p*-value <0.05 was considered statistically significant. Survival analyses were performed using MedCalc for Windows, version 15.0 (MedCalc Software, Ostend, Belgium). Data processing for the VEGF/VEGFR analyses in gastric cancer tissue was completed using R statistical environment version 3.6.2 and figures were generated using GraphPad Prism version 6.07.

## 5. Conclusions

The limitation of this study is the relative sample size, so our findings warrant further investigations to confirm the association between transcriptionally inactive *TP53* missense mutations and improved clinical outcomes of patients with metastatic gastric adenocarcinoma who received anti-VEGFR2 plus Paclitaxel systemic therapy. From a clinical perspective, the *TP53_RTAS_* mutational analysis might improve the identification of patients who are likely to have the greatest benefit from Ramucirumab therapy. Ramucirumab and chemotherapy failed to achieve significant survival advantages in a randomized phase III study when adopted as a first-line therapy for metastatic gastric cancer [35]. In the overall treatment strategy for the metastatic disease, the selection of patients according to *TP53_RTAS_* mutational status represents a promising model to tailor treatment choices and improve clinical outcomes. In addition, *TP53_RTAS_* analysis could be evaluated in patients with metastatic adenocarcinomas in other solid tumors with frequent *TP53* mutations and where anti-VEGF therapy is commonly employed.

## Figures and Tables

**Figure 1 cancers-12-02049-f001:**
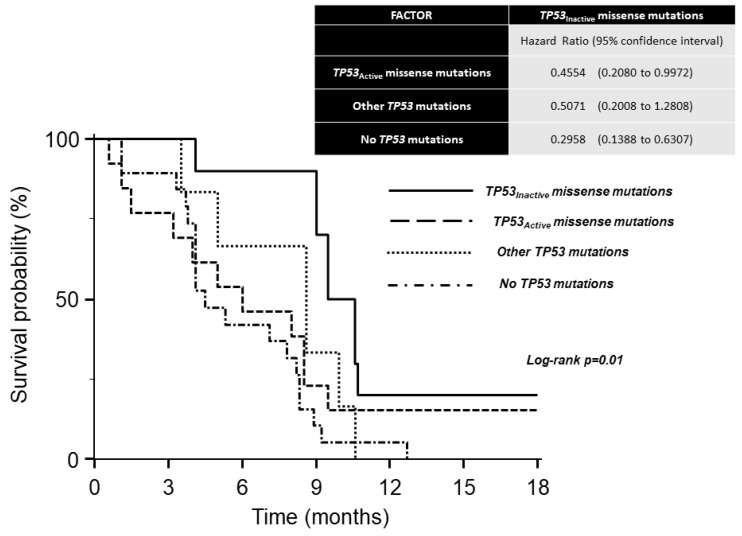
Kaplan–Meier survival curves of Ramucirumab/Paclitaxel second-line therapy in 48 patients with metastatic gastric cancer.

**Figure 2 cancers-12-02049-f002:**
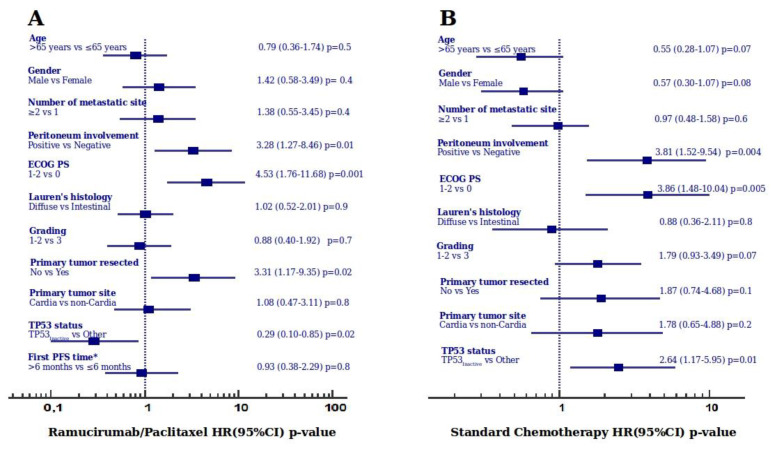
Results of the multivariate model analysis for overall survival in the Ramucirumab/Paclitaxel (**A**) and Standard Chemotherapy (**B**) treatment groups. Abbreviations: HR, Hazard Ratio; CI, confidence interval; ECOG PS, Eastern Cooperative Group Performance Status; PFS, progression-free survival. First PFS time* is a variable for second-line Ramucirumab/Paclitaxel therapy only.

**Figure 3 cancers-12-02049-f003:**
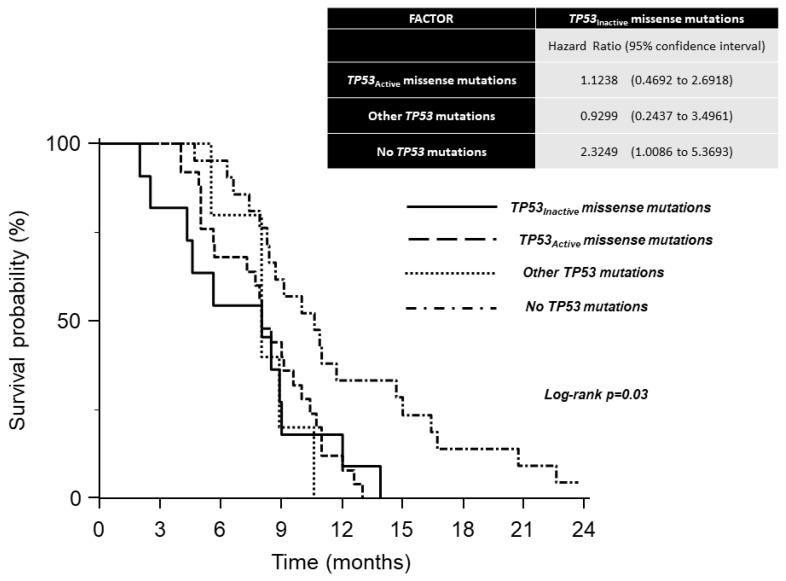
Kaplan–Meier survival curves of first-line combination chemotherapy in 62 patients with metastatic gastric cancer.

**Figure 4 cancers-12-02049-f004:**
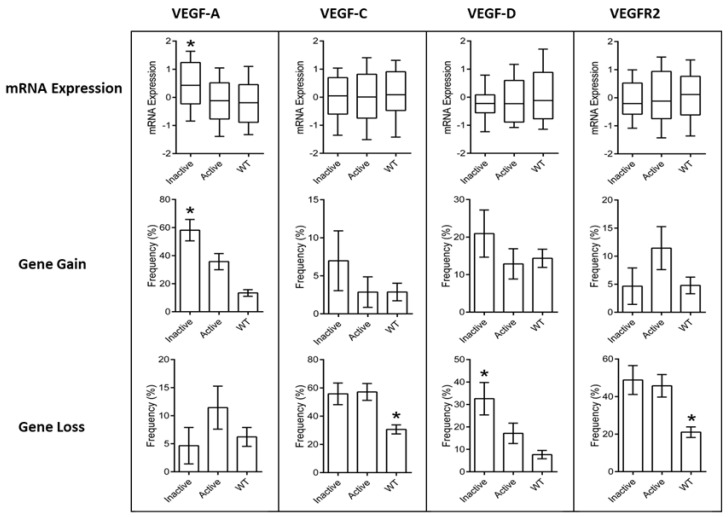
Plots of vascular endothelial growth factor (VEGF)/VEGF receptor 2 (VEGFR2) analysis in gastric cancer tissues. Data were collected from the TCGA PanCancer Atlas. * indicates statistically significant differences between groups as described in the text.

**Table 1 cancers-12-02049-t001:** Description of the tumor protein p53 (*TP53*) mutations detected in 70 patients.

Mutation	Amino Acid Change	Effect	RTAS	Functional Classification	Hg19 Coordinates	Therapy Group
G > T	G245C	missense mutation	0	Inactive	7577548	R/P-SC
G > A	M246I	missense mutation	0	Inactive	7577543	R/P
C > T	R248W	missense mutation	0	Inactive	7577539	R/P-SC
C > T	R282W	missense mutation	0	Inactive	7577094	R/P ^2^-SC
G > A	R283H	missense mutation	0	Inactive	7577090	R/P ^3^
C > T	T304I	missense mutation	0	Inactive	7577027	R/P-SC
G > A	G244D	missense mutation	0.2	Inactive	7577550	R/P-SC ^2^
C > T	R273C	missense mutation	0.4	Inactive	7577121	SC3 ^5^
G > A	V216M	missense mutation	1.2	Active	7578203	SC
C > T	P151S	missense mutation	5.2	Active	7578479	SC
G > A	R175H	missense mutation	9.2	Active	7578406	R/P-SC
T > C	I195T	missense mutation	11.4	Active	7578265	SC
C > G	P177R	missense mutation	12.0	Active	7578400	R/P
C > T	L194F	missense mutation	12.0	Active	7578269	SC
C > T	S260F	missense mutation	12.6	Active	7577502	SC
G > A	G105S	missense mutation	15.0	Active	7579374	SC
C > T	H214Y	missense mutation	20.9	Active	7578209	SC
C > T	H179Y	missense mutation	22	Active	7578395	R/P
G > A	E180K	missense mutation	22.8	Active	7578392	R/P
C > T	P177S	missense mutation	26.9	Active	7578401	SC
G > A	R282Q	missense mutation	30.5	Active	7577093	R/P
C > T	P190S	missense mutation	32.0	Active	7578281	SC
C > T	R181C	missense mutation	32.4	Active	7578389	R/P
G > A	D228N	missense mutation	40.7	Active	7577599	SC
G > A	C229Y	missense mutation	69.3	Active	7577595	SC
C > T	R175C	missense mutation	72.5	Active	7578407	R/P
C > T	L252F	missense mutation	76.7	Active	7577527	SC
G > A	R379H	missense mutation	77.8	Active	7572974	SC
C > T	H115Y	missense mutation	81.1	Active	7679344	R/P
G > A	G356R	missense mutation	88.3	Active	7573961	SC
C > T	S116F	missense mutation	90.7	Active	7579340	SC
G > A	V225I	missense mutation	91.7	Active	7577608	R/P
G > A	A353T	missense mutation	96.9	Active	7573970	SC
C > T	L383F	missense mutation	97.5	Active	7572962	R/P
C > T	S90F	missense mutation	99.2	Active	7579418	SC
G > A	R174K	missense mutation	102.0	Active	7578409	SC
C > T	P222L	missense mutation	102.9	Active	7578184	R/P
G > A	E294K	missense mutation	107.7	Active	7577058	SC
G > A	S261N	missense mutation	108.0	Active	7577499	SC
C > T	S314F	missense mutation	110.0	Active	7576905	SC
G > A	V217M	missense mutation	116.0	Active	7578200	SC
G > A	G226D	missense mutation	120.1	Active	7577604	R/P
C > T	R290C	missense mutation	134.2	Active	7577070	SC
C > T	T329I	missense mutation	138.6	Active	7576860	SC
C > T	T312I	missense mutation	139.8	Active	7576911	R/P
G > A	A307T	missense mutation	142.7	Active	7577019	SC
C > T	P309S	missense mutation	151.2	Active	7576920	R/P
C > T	R196 *	nonsense mutation	-	Other	7578263	SC
C > T	Q192 *	nonsense mutation	-	Other	7578275	SC
C > T	R342 *	nonsense mutation	-	Other	7574003	R/P
C > T	Q317 *	nonsense mutation	-	Other	7576897	SC
C > T	R306 *	nonsense mutation	-	Other	7577002	R/P
C > T	Q165 *	nonsense mutation	-	Other	7578437	SC
C > G	Y107 *	nonsense mutation	-	Other	7579366	R/P
GTC > GT	L93X	reading frameshift	-	Other	7579408	R/P
tGCCCCCac > tTCCCCCCac	CPH176-178FPPX	reading frameshift	-	Other	7578397-403	SC
GCCCCCTCC > gCCCCTCcc	APS88-90VPS	reading frameshift	-	Other	7579419-424	R/P
AGA > A	R209X	reading frameshift	-	Other	7578221-223	SC
CCT > -	P190-	inframe deletion	-	Other	75782780-281	R/P
G > T	-	acceptor intron 8	-	Other	7576927	R/P
G > A	-	acceptor intron 9	-	Other	7576852	SC

Abbreviations: RTAS, residual transcriptional activity score; SC, standard chemotherapy; R/P, Ramucirumab/Paclitaxel; hg19, Genome Reference Consortium Human Build 37 (GRCh37) coordinates; Legend: ^2^ mutation in two cases; ^3^ mutation in three cases; ^5^ mutation in five cases; * stop codon.

**Table 2 cancers-12-02049-t002:** Characteristics and distribution of the 110 patients according to treatments and *TP53* status.

Number of Patients (%)							
Variable	Ramucirumab/Paclitaxel		Standard Chemotherapy		Total		*p*-Value
	TP53 wt	TP53 mut	TP53 wt	TP53 mut	TP53 wt	TP53 mut	
Age							
>65 years	12 (63.2)	15 (51.7)	11 (52.4)	20 (48.8)	23 (57.5)	35 (50)	0.5
≤65 years	7 (36.8)	14 (48.3)	10 (47.6)	21 (51.2)	17 (42.5)	35 (50)	
Gender							
Male	11 (57.9)	16 (55.2)	15 (71.4)	23 (56.1)	26 (65)	39 (55.7)	0.4
Female	8 (42.1)	13 (44.8)	6 (28.6)	18 (43.9)	14 (35)	31 (44.3)	
Grading							
1–2	16 (84.2)	20 (68.9)	14 (66.6)	24 (58.5)	30 (75)	44 (62.8)	0.2
3	3 (15.8)	9 (31.1)	7 (33.4)	17 (41.5)	10 (25)	26 (37.2)	
Peritoneum involvement							
Positive	10 (52.6)	16 (55.2)	3 (14.3)	14 (34.2)	13 (42.5)	30 (42.8)	0.4
Negative	9 (47.4)	13 (44.8)	18 (85.7)	27 (65.8)	27 (67.5)	40 (57.2)	
ECOG PS							
0	9 (47.3)	18 (62.1)	18 (85.7)	24 (58.5)	27 (67.5)	42 (60)	0.5
1–2	10 (52.7)	11 (37.9)	3 (14.3)	17 (41.5)	13 (42.5)	28 (40)	
Lauren’s histology							
Intestinal	12 (63.2)	23 (79.3)	11 (52.4)	32 (78)	23 (57.6)	55 (78.5)	0.02
Diffuse	7 (36.8)	6 (20.7)	10 (47.6)	9 (22)	17 (42.5)	15 (21.5)	
Grading							
1–2	10 (52.6)	19 (65.5)	12 (57.1)	31 (75.6)	22 (55)	50 (71.4)	0.09
3	9 (47.4)	10 (34.5)	9 (42.9)	10 (24.4)	18 (45)	20 (28.6)	
Primary tumor resected							
Yes	12 (63.2)	9 (31.1)	10 (52.4)	18 (43.9)	22 (55)	27 (38.5)	0.1
No	7 (36.8)	20 (68.9)	11 (47.6)	23 (56.1)	18 (45)	43 (61.5)	
Primary tumor site							
Cardia	7 (36.8)	11 (37.9)	9 (42.9)	15 (36.5)	16 (40)	26 (37.1)	0.8
non-cardia	12 (63.2)	18 (62.1)	12 (57.1)	26 (63.5)	24 (60)	44 (62.9)	


Abbreviations: wt, wild-type; mut, mutated; ECOG PS, Eastern Cooperative Group Performance Status.

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
