# Peer review of "TP53 Mutation Analysis in Gastric Cancer and Clinical Outcomes of Patients with Metastatic Disease Treated with Ramucirumab/Paclitaxel or Standard Chemotherapy"

_cancers, 2020, doi:10.3390/cancers12082049_

Round 1

Reviewer 1 Report

Summary:

The authors have taken a very interesting approach to demonstrate that not all TP53 mutation types result in the same outcomes. Excitingly, they have identified potentially clinically relevant mutation types that could be easily screened for, and utilized in the consideration of treatment options. The manuscript could be significantly improved by providing a more detailed description of the methodologies used however, particularly when it comes to derivation and utilization of the RTAS scores.  

Major concerns:

  1. The major findings of the work are based on a combination of methodologies from two previously funded manuscripts, but the approach taken to utilize the findings/methods from these works are poorly described. For example, the RTAS reported for the V216M p53 mutant in one of the cited references (Fischer NW, Prodeus A, Gariépy J. Survival in males with glioma and gastric adenocarcinoma correlates with mutant p53 residual transcriptional activity. JCI Insight. 2018;3(15):e121364. Published 2018 Aug 9. doi:10.1172/jci.insight.121364) is 1.2, but in the current manuscript it is reported as 2.7. Where did the discrepancy come from? If the authors used a different approach, then it needs to be described.
    1. It would also be incredibly helpful to just summarize how the RTAS scores were derived rather than forcing the reader to find and read two other papers to understand the methodology.
  2. RTAS scores are provided in table two with zero context as to their meaning. It is not until section 4.5 that only those with an RTAS score of < 1 are considered inactive, while all others are “active.” Based on the way the analyses were done, some sort of designation on the table indicating “active” versus “inactive” may be a more appropriate way to segregate RTAS mutation types as all mutants with RTAS scores ranging from 2.7-151.1 were considered equally “active.”

Minor concerns:

  1. In section 2.1, the sentence says 70 out of 100 patients, but the number should be out of 110 patients.
  2. Sections 2.3 and 2.4 refer to PRs and CRs, but these abbreviations are not defined until section 4.5.
  3. Similarly, section 2.5 refers to CNAs, which are not defined until section 4.4.

Author Response

Reviewer #1

Major concerns:

1. The major findings of the work are based on a combination of methodologies from two previously funded manuscripts, but the approach taken to utilize the findings/methods from these works are poorly described. For example, the RTAS reported for the V216M p53 mutant in one of the cited references (Fischer NW, Prodeus A, Gariépy J. Survival in males with glioma and gastric adenocarcinoma correlates with mutant p53 residual transcriptional activity. JCI Insight. 2018;3(15):e121364. Published 2018 Aug 9. doi:10.1172/jci.insight.121364) is 1.2, but in the current manuscript it is reported as 2.7. Where did the discrepancy come from? If the authors used a different approach, then it needs to be described.

The RTAS for V216M was a typing error and should be written as 1.2, as the reviewer has stated. Other discrepancies in RTAS values were due to number rounding inconsistencies. The values have been adjusted in Table 1 for consistency to match previous reports (lane 152, Table 2). These adjustments do not affect the results, classification of mutations, or conclusions.

1. It would also be incredibly helpful to just summarize how the RTAS scores were derived rather than forcing the reader to find and read two other papers to understand the methodology.

A detailed description of the RTAS score derivation has been added to the Methods section 4.3 for clarification (lane 425).

2. RTAS scores are provided in table two with zero context as to their meaning. It is not until section 4.5 that only those with an RTAS score of < 1 are considered inactive, while all others are “active.” Based on the way the analyses were done, some sort of designation on the table indicating “active” versus “inactive” may be a more appropriate way to segregate RTAS mutation types as all mutants with RTAS scores ranging from 2.7-151.1 were considered equally “active.”

An additional column labelled “Functional Classification” has been added to Table 1 to denote the TP53 mutations as “active”, “inactive”, or “other”.

Minor concerns:

1. In section 2.1, the sentence says 70 out of 100 patients, but the number should be out of 110 patients.

Thank you. We corrected the typing mistake (lane 145).

2.Sections 2.3 and 2.4 refer to PRs and CRs, but these abbreviations are not defined until section 4.5.

We substituted the abbreviations RR, CR, PR with the full name. Lanes 182, 183, 185, 213, 214, 217, 220, 458, 459, 461, 462.

3. Similarly, section 2.5 refers to CNAs, which are not defined until section 4.4.

We changed the abbreviation in copy number alterations (lane 241).

Reviewer 2 Report

The manuscript under review is well-written and the study objective is definitely of scientific interest. However, the study design is limited by its small sample size - this should be discussed as a limitation in the discussion section. Another point of criticism is that 22 authors are listed which does not seem justified in light of the study design and sample size - please check whether the contributions of the persons listed as authors really meet the authorship requirements (usually, no more than 10 authors should be listed). I actually doubt it.

Otherwise, I think the manuscript only needs a minor final language check. 

Conclusion/recommendation: minor revision (authors and contributions, detailed discussion of the limitations of the study in the discussion).

Author Response

The manuscript under review is well-written and the study objective is definitely of scientific interest. However, the study design is limited by its small sample size - this should be discussed as a limitation in the discussion section. Another point of criticism is that 22 authors are listed which does not seem justified in light of the study design and sample size - please check whether the contributions of the persons listed as authors really meet the authorship requirements (usually, no more than 10 authors should be listed). I actually doubt it.

Relatively to the criticism about too many authors, as the reviewer can see, we stated under Author contributions section “who made what”. Most of the authors in fact contributed to collect samples (pathologists and oncologists). Gastric cancer tissues from patients treated with Ramucirumab/Paclitaxel suitable for the study are not easy to collect as it seems. Under Conclusion section we added a sentence to state the limitation of the study, relatively to sample size.

we checked the language

Reviewer 3 Report

TP53 mutation analysis in gastric cancer and clinical outcomes of patients with metastatic disease treated with Ramucirumab-Paclitaxel or standard chemotherapy

This study by Graziano et al. is significant because it demonstrates better outcome of a second-line therapy (Ramucirumab+Paclitaxel) versus standard chemotherapy (5-Flurouracil + Cisplatin/Oxaliplatin) for a sub-cohort of gastric cancer patients with transcriptionally inactive TP53 mutations. The authors show that TP53inactive patients have upregulated VEGF-A expression, respond best to Ramucirumab-Paclitaxel while the same TP53inactive patients group respond worst to standard chemotherapy. This finding is important because it creates the opportunity to better target gastric cancers with transcriptionally inactive TP53 mutations that tend to be more angiogenic. The findings are important and interesting, and the study is well done with appropriate controls. However, to improve the manuscript there are a few major points that need to be addressed. If those points can be addressed, the manuscript would be suitable for publication in Cancers.

Major points:

  • I do not see that the RNAseq data has been publicly deposited. The authors need to deposit the data in Gene Expression Omnibus (https://www.ncbi.nlm.nih.gov/geo/info/submission.html) or other public database along with the patient metadata. This is essential so that other researchers can replicate this analysis and/or conduct new analysis with the data.
  • The data in Table 2 would be more accessible to the reader if the authors included stacked bar plots with the data from Table 2. For example, see https://www.r-graph-gallery.com/stacked-barplot.html
  • I recommend replacing Table 3 with a plot like the one in the following link: https://www.jmp.com/support/downloads/JMPC71_documentation/Content/JMPCUserGuide/GR_C_0013.htm#:~:text=A%20hazard%20ratio%20event%20plot,of%20subjects%20in%20a%20study.&text=Each%20bar%20represents%20the%20hazard,caused%20by%20the%20second%20treatment.
  • I also think that the authors should make new bar plots with the RR, PR and CR numbers that are discussed in sections 2.3 and 2.4

Minor points:

  • Authors should in detail describe how they calculated residual transcriptional activity score (RTAS) in the materials and methods.

Author Response

Major points:

I do not see that the RNAseq data has been publicly deposited. The authors need to deposit the data in Gene Expression (https://www.ncbi.nlm.nih.gov/geo/info/submission.html) or other public database along with the patient metadata. This is essential so that other researchers can replicate this analysis and/or conduct new analysis with the data.

The RNAseq data was collected from the TCGA PanCancer Atlas (https://www.cancer.gov/tcga). For clarity, we have added a statement and the website link in the Materials section 4.4 (lane 436). In addition, we have added a statement in Figure 4 (lane 260) (lane 260) to clarify the source of the data that was collected for analysis. We don’t think that we need to deposit patients metadata.

The data in Table 2 would be more accessible to the reader if the authors included stacked bar plots with the data from Table 2.

We tried to create a stacked bar plot with data from Table 2 but we think that it is better to leave the Table 2.

I recommend replacing Table 3 with a plot.

Thank you for his suggestion. We substituted Table 3 with the new Figure 2.

I also think that the authors should make new bar plots with the RR, PR and CR numbers that are discussed in sections 2.3 and 2.4

We think that the sections 2.3 and 2.4 are clearly described and are well represented by Kaplan-Meier curves (Figures 1 and 3).

Minor points: Authors should in detail describe how they calculated residual transcriptional activity score (RTAS) in the materials and methods.

A detailed description of the RTAS score derivation has been added to the Methods section 4.3 (lane 425) for clarification.